# Underweight Is Associated with a Higher Risk of Acute Pancreatitis in Type 2 Diabetes: A Nationwide Cohort Study

**DOI:** 10.3390/jcm11195641

**Published:** 2022-09-25

**Authors:** Young Hoon Choi, Kyung-Do Han, In Rae Cho, In Seok Lee, Ji Kon Ryu, Yong-Tae Kim, Kwang Hyun Chung, Sang Hyub Lee

**Affiliations:** 1Department of Internal Medicine, College of Medicine, The Catholic University of Korea, Seoul 06591, Korea; 2Department of Statistics and Actuarial Science, Soongsil University, Seoul 06978, Korea; 3Department of Internal Medicine and Liver Research Institute, Seoul National University Hospital, Seoul National University College of Medicine, Seoul 03080, Korea; 4Division of Gastroenterology, Department of Internal Medicine, Uijeongbu Eulji Medical Center, Eulji University School of Medicine, Uijeongbu-si 11759, Korea

**Keywords:** pancreatitis, diabetes mellitus, type 2, body mass index, cohort studies, nationwide studies

## Abstract

Type 2 diabetes is known as a risk factor for acute pancreatitis, but the risk of acute pancreatitis according to glycemic status and body mass index (BMI) has remained unknown. Therefore, we aim to investigate the risk of acute pancreatitis according to BMI and glycemic status. We included 3,912,496 subjects from the Korean National Health Insurance System cohort who underwent the National Health Screening program in 2009. Each subject’s clinical course was examined through follow-ups until December 2018. BMI and glycemic status were each categorized into five groups. Hazard ratios (HRs) of acute pancreatitis according to BMI and glycemic status were calculated. The adjusted HRs of acute pancreatitis were the highest in the underweight group (BMI < 18.5) in all five glycemic status categories. The HR of acute pancreatitis in the underweight group increased as the glycemic status worsened, excluding the category of diabetes for more than five years (HR 1.381 for normal fasting glucose; 1.805 for impaired fasting glucose; 2.332 for new-onset diabetes; 4.51 for diabetes duration <5 years; 4.135 for diabetes duration ≥5 years). We found that the risk of acute pancreatitis was further increased in the underweight group, depending on the status and duration of type 2 diabetes.

## 1. Introduction

Acute pancreatitis, a sudden inflammatory disorder of the pancreas, is one of the most common gastrointestinal diseases requiring hospitalization [1]. The reported incidence of acute pancreatitis ranges from four to over 100 per 100,000 person-years and has been increasing worldwide [2,3]. Some clinical factors that are known as risk factors for acute pancreatitis, such as gallstones and obesity, are significantly associated with type 2 diabetes [4,5]. According to several studies, it is estimated that patients with type 2 diabetes have a higher risk of acute pancreatitis by 1.5 to 1.9 times than the general population, even after adjusting for those clinical factors [6,7,8]. A high body mass index (BMI) is also known to increase the risk of acute pancreatitis in several reports [9,10]. Therefore, it may be inferred that the risk of acute pancreatitis is higher in patients with type 2 diabetes with a high BMI, but no studies have been conducted to evaluate the risk of acute pancreatitis in relation to both diabetes status and BMI. There are studies that have suggested otherwise. A study by Coleman et al. reported a contrasting result of higher frequency of pancreatitis history in diabetic patients with a low BMI than in those with a high BMI, though limitations in deriving a causal relationship exist, due to the lack of information on the onset of pancreatitis and diabetes [11]. In addition, although not a study on diabetic patients, a recent study with a large general population reported that not only high but also a low BMI raises the risk of acute pancreatitis [12]. Therefore, this study aims to evaluate the risk of acute pancreatitis in relation to type 2 diabetes status and BMI, using a large-scale database from the Korean National Health Insurance Service (NHIS). 

## 2. Materials and Methods

### 2.1. Data Source and Study Subjects

This study used the national health checkup database from the Korean NHIS, a single compulsory insurance system that covers 97% of the Korean population and provides a biennial national health checkup program [13]. The national health screening (NHS) program, targeting all insured people over the age of 40 and all employees over the age of 20, includes anthropometric measurements, laboratory tests, and questionnaires about lifestyle behaviors, such as smoking, drinking, and physical activity. In addition to such information from the NHS program, the NHIS database contains demographic data, such as age, sex, and income level, clinic usage data, diagnostic codes in the form of International Statistical Classification of Diseases and Related Health Problems 10th Revision (ICD-10), and medical claim data. 

Out of approximately ten million individuals that underwent the NHS program in 2009, 4,238,822 subjects, approximately 40%, were randomly selected for this study. We excluded subjects under the age of 20 (n = 4481), those with a history of acute pancreatitis before 2009 (n = 63,939), those diagnosed with acute pancreatitis within a year before 2009 (n = 12,028) and those with missing data (n = 245,878). Finally, a total of 3,912,496 subjects were enrolled in this study and followed up until 31 December 2018.

This study was conducted in accordance with the Declaration of Helsinki and approved by the institutional review board (IRB) of Soongsil University (IRB No. SSU-202007-HR-236-01). Since the NHIS database provides de-identified data publicly available for research, the IRB waived the requirement for informed consent.

### 2.2. Definitions of Acute Pancreatitis and Chronic Diseases

In this particular study, acute pancreatitis was defined as a case in which ICD-10 code K85 labeled the primary diagnosis in hospitalized patients. This definition of acute pancreatitis showed a positive predictive value of 82–98% for acute pancreatitis in previous studies [14,15]. Hypertension was defined as a systolic blood pressure of 140 mmHg or higher, a diastolic blood pressure of 90 mmHg or higher, or a diagnosis of ICD-10 codes I10-I13 and I15 and a prescription for antihypertensive drugs at least once a year. Dyslipidemia was defined as a total cholesterol level of 240 mg/dL or higher or lipid-lowering agents prescribed at least once a year according to ICD-10 code E78. Chronic kidney disease (CKD) was defined as an estimated glomerular filtration rate (GFR) of less than 60 mL/min/1.73 m^2^, calculated using the Modification of Diet in Renal Disease (MDRD) equation [16]. Chronic pancreatitis was defined as a primary diagnosis of ICD-10 code K86.0 or K86.1.

### 2.3. Assessment of Obesity and Glycemic Status

Body mass index (BMI) was classified into five groups according to the World Health Organization (WHO) recommendations for Asians: underweight (BMI < 18.5 kg/m^2^), normal range (BMI 18.5–22.9 kg/m^2^), overweight (BMI 23–24.9 kg/m^2^), obese stage I (BMI 25–29.9 kg/m^2^), and obese stage II (BMI ≥ 30 kg/m^2^) [17]. The constant BMI maintenance group was defined as a set of subjects who had been in the same BMI classification for a year from the date of NHS program of 2009.

The glycemic status of subjects was categorized into five groups as follows: no diabetes (fasting plasma glucose [FPG] level < 100 mg/dL), impaired fasting glucose (IFG) (FPG level 100–125 mg/dL), new-onset type 2 diabetes, type 2 diabetes with a duration of less than five years, and type 2 diabetes with a duration of more than five years. Patients with type 2 diabetes were defined as subjects with ICD-10 codes E11–14 and prescription records of antidiabetic medications, or with an FPG level of ≥126 mg/dL [18]. New-onset diabetes was defined as diabetes first diagnosed in the 2009 National Health Checkup, and the classification according to the 5-year prevalence period was also based on the diabetes prevalence period as of 2009.

### 2.4. Lifestyle Behaviors and Economic Status

Based on their smoking statuses, subjects involved were classified as non-smokers, former smokers, and current smokers. Drinking behaviors also classified subjects as non-drinkers, mild-drinkers (alcohol intake < 30g/day), and heavy-drinkers (alcohol intake ≥ 30g/day). Moderate exercise for more than 30 min at least five times a week or vigorous exercise for more than 20 min at least three times a week was defined as regular physical activity. Income levels were divided into quartiles of which the lower quartile (under 25%) was defined as low-income.

### 2.5. Statistical Analysis

Categorical data were shown as numbers (%), and continuous data were shown as mean ± SD or geometric mean with 95% confidence interval (CI). One-way analysis of variance and chi-squared test were used appropriately to compare baseline characteristics. We used Cox proportional-hazards model to obtain a hazard ratio (HR) with 95% CI for acute pancreatitis according to BMI category and glycemic status. Model 1 was non-adjusted. Model 2 was adjusted for age and sex. Model 3 was further adjusted for alcohol consumption, smoking, type 2 diabetes, insulin administration, hypertension, dyslipidemia, CKD, chronic pancreatitis, regular physical activity, and low income. The incidence rate (IR) of acute pancreatitis was calculated and presented in units of 1000 person-years. A P-value of less than 0.05 was considered statistically significant. Statistical analyses were performed using SAS 9.4 (SAS Institute Inc., Cary, NC, USA). 

## 3. Results

### 3.1. Baseline Characteristics

During a median follow-up period of 8.31 (interquartile range, 8.11–8.57) years, 8933 of 3,912,496 subjects were diagnosed with acute pancreatitis. Baseline characteristics of the subjects are classified by the presence or absence of a low BMI (<18.5 kg/m^2^) and diabetes, as summarized in Table 1. In subjects with diabetes and a relatively a high BMI (≥18.5 kg/m^2^), the proportions of males, heavy drinkers, hypertensive patients, dyslipidemia patients, and subjects with regular physical activity were higher. Underweight subjects with diabetes were older and had a higher prevalence of CKD.

### 3.2. Risk of Acute Pancreatitis for Each BMI Group Depending on the Presence or Absence of Type 2 Diabetes

The HR of acute pancreatitis was higher in the underweight category (HR 1.408; 95% CI 1.261–1.573) than in other BMI categories. In particular, the HR in the underweight group with diabetes (HR 1.836; 95% CI 1.347–2.503) was the highest. The overweight group a showed lower HR of acute pancreatitis (HR 0.844; 95% CI 0.798–0.893) than those of other BMI groups. The overweight group with diabetes showed the lowest HR (HR 0.734; 95% CI 0.639–0.843). As BMI increased from obese stage I to II, the HR of acute pancreatitis also increased. Amongst those in obese stage II, all subjects in total and subjects without diabetes had HRs of 1.15 (95% CI 1.036–1.275) and 1.235 (95% CI 1.097–1.392), respectively, ratios higher than those in the normal range BMI category. However, since the increase in HR was relatively small in subjects with diabetes, the HR was 0.831 (95% CI 0.67–1.03) even in obese stage II. As a result, the HR of acute pancreatitis according to BMI was U-shaped in all subjects and subjects without diabetes, whereas, in subjects with diabetes, only in the underweight category was the HR higher than that of the normal range BMI category (Figure 1, Appendix A).

### 3.3. Risk of Acute Pancreatitis According to Glycemic Status and BMI Category

The IR (per 1000 person-years) and HR of acute pancreatitis according to five glycemic statuses, each further categorized by five BMI ranges are shown in Figure 2 (Appendix A). In the normal FPG group, the IR was highest in the obese stage II (IR 0.293). However, in the IFG or diabetes groups, the IR was highest in the underweight category and increased as glycemic status worsened from IFG to type 2 diabetes of less than five years (IR 0.466 for IFG; 0.858 for new-onset type 2 diabetes; 1.78 for type 2 diabetes <5 years). In the group with type 2 diabetes of more than five years, IR did not increase further but showed a higher value than that in the group of new-onset type 2 diabetes (IR 1.634 for type 2 diabetes ≥5 years).

The underweight category in all five glycemic statuses had the highest HR of acute pancreatitis. Similar to the pattern found in IRs, the increase of overall HRs correlated the worsening of glycemic status and duration of diabetes, as exhibited in groups from IFG to diabetes of less than five years. (HR 1.381 for normal FPG; 1.805 for IFG; 2.332 for new-onset type 2 diabetes; 4.51 for type 2 diabetes <5 years). The HR in diabetes of more than five years (HR 4.135 for type 2 diabetes ≥5 years) was not higher than that in diabetes of less than five years, but higher than that in new-onset type 2 diabetes. 

### 3.4. Subgroup Analysis with Constant BMI

The adjusted HRs of acute pancreatitis in subjects with constant BMI according to the presence or absence of type 2 diabetes are shown in Figure 3 (Appendix A). We used the 2010 glycemic status of subjects in this analysis. There were a total of 1,224,892 subjects with constant BMI: 1,162,479 subjects without and 62,413 subjects with type 2 diabetes. Subjects with constant BMI of less than 18.5 showed the highest HR of acute pancreatitis in both with and without type 2 diabetes (HR 1.674 for non-diabetes and HR 1.311 for type 2 diabetes). 

## 4. Discussion

To the best of our knowledge, this is the first study to establish the effect of BMI and glycemic status on the risk of developing acute pancreatitis. We examined 3,912,496 subjects from the NHIS cohort over a median follow-up period of 8.3 years. In our study, the risk of acute pancreatitis increased in the underweight or obese groups, and the HR of acute pancreatitis was further increased in the underweight group. Notably, the risk of acute pancreatitis in underweight subjects increased in correlation with the presence and duration of type 2 diabetes. Such a pattern was also observed in the subgroup analysis for patients with constant BMI.

Several studies on the risk of acute pancreatitis according to BMI in the general population have been conducted. The results are conflicting: in most studies, a high BMI increased the risk of acute pancreatitis, with no significant effect found in a low BMI [9,10]. However, a recently published study in Korea demonstrated an increased risk of acute pancreatitis in not only high but also low BMI groups, similar to our findings [12]. There are several possible explanations for such varying results amongst studies. First, previous studies may lack statistical significance due to the relatively small number of subjects. Considering that the aforementioned study in Korea has the largest sample size of approximately half a million, amongst the previous studies, and that our study, which involves a larger sample of approximately four million, has correlating results, it is plausible that the effect of a low BMI on the risk of acute pancreatitis holds statistical significance [12]. Secondly, given that a low BMI is correlated with increased HR of acute pancreatitis in our study, other studies define a low BMI as less than 20, a comparatively lenient criteria that may include individuals with relatively higher BMI, which may have factored in decreasing the HR [19,20]. Thirdly, since most previous studies have been conducted in western countries, differences due to ethnicity may have varied study results. In the study of Hansen et al. conducted in Denmark, the low BMI criterion was set at 18.5, the same as in our study, and the adjusted clinical factors were similar to our study, but there was no significant increase in the risk of pancreatitis in the underweight group [10]. This suggests that the differences in the study results may be due to differences according to ethnicity. 

It has been well known from previous studies that type 2 diabetes increases the risk of acute pancreatitis [7,21]. However, the relationship between IFG or new-onset diabetes and the risk of acute pancreatitis lacks research. In our study, the risk of acute pancreatitis increased with the presence of IFG and new-onset diabetes. 

Although several studies have been conducted on the relationship between BMI and acute pancreatitis in the general population and the association between diabetes and acute pancreatitis, the risk of acute pancreatitis according to diabetes status and BMI remains unknown. According to a study on diabetes, BMI, and acute pancreatitis, a history of pancreatitis was more frequent in lean diabetic patients than in obese diabetic patients [11]. However, there was no information on the time sequence of onset of diabetes and acute pancreatitis in this study; thus, it could not be concluded that a low BMI was a risk factor for pancreatitis in diabetic patients. In our study, in order to clarify the time sequence of the onset of diabetes and acute pancreatitis, patients who had a history of acute pancreatitis before participation in our study were excluded. In addition, in order to understand the effect of the degree of diabetes on the risk of acute pancreatitis, glycemic status was divided into five categories. As a result, we found that the overall risk of acute pancreatitis increased as glycemic status worsened and diabetes continued for a longer period, from normal FPG to diabetes of less than five years, and such a trend was particularly pronounced in underweight subjects. The cause of a marked increase in the risk of acute pancreatitis throughout diabetes progression in underweight subjects is not clear, but some explanations can be suggested. Poor glycemic control in the underweight group may serve as an explanation. Though we could not evaluate the patients’ long-term glycemic control, since no information on HbA1c was provided in the database, baseline FPG level was higher in the diabetic patients with a BMI less than 18.5 than in patients with a BMI 18.5 or higher. Several studies have also reported poor glycemic control in lean diabetes patients [22,23]. A study by Mohan et al. involving 9783 type 2 diabetes patients showed that FPG, postprandial plasma glucose and HbA1c levels were higher in patients with a BMI less than 18.5 than in patients with a BMI 18.5 or higher [23]. 

Our study also reported that HR of acute pancreatitis no longer increased in patients with diabetes for more than five years. It is difficult to conclude that the HR in diabetes of more than five years deviated from the overall trend: while the ratio is lower than that of diabetes of less than five years, it still remains higher than that of newly-onset diabetes or IFG. As some reports suggest that statin drugs commonly taken in diabetic patients lower the risk of pancreatitis, the slight decrease in HR of acute pancreatitis in diabetes of more than five years may be explained by the effect of drugs, such as statin; however, further studies are necessary [24]. 

There are several limitations in this study. First, we adjusted for several confounding factors but could not adjust for gallstone disease, one of the most common causes of acute pancreatitis. However, as various reports suggest that gallstone is associated with a high BMI, it is unlikely that the high HR of acute pancreatitis in the low BMI group was associated with gallstone [12,25]. Secondly, the effect of the drugs in patients was not taken into consideration. Thirdly, because the cohort used in our study was provided by the Korean NHIS, the ethnicity of the study subjects was mostly limited to Asians. Fourth, the operational definitions of covariables in this study are estimated to be relatively close to the actual data because they were based on the Korean NHIS claim data, which is strictly managed by the government, but there are difficult to quantify parts as well as those where it was difficult to confirm accuracy. To overcome this, a system that links claim data and real-world hospital data is needed, but in our study data based on such a system could not be secured.

In summary, our study found an increased risk of acute pancreatitis with the glycemic status and duration of type 2 diabetes, especially in patients with a low BMI. Based on our findings, physicians need to be aware that a low BMI as well as a high BMI can increase the risk of acute pancreatitis when managing patients with type 2 diabetes.

## Figures and Tables

**Figure 1 jcm-11-05641-f001:**
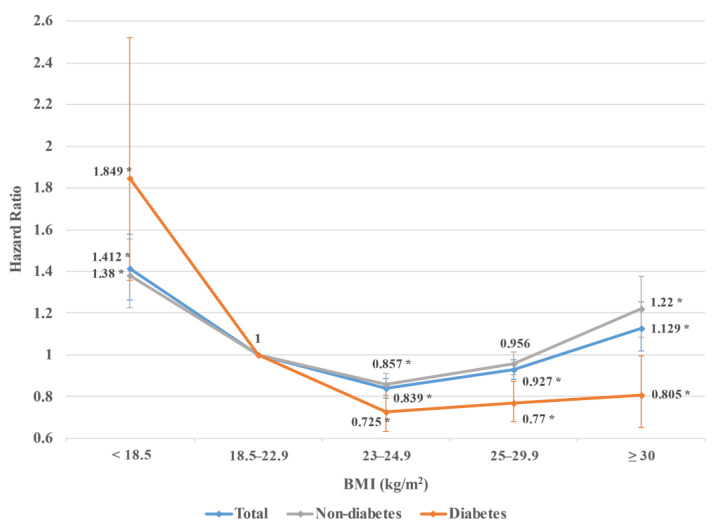
Adjusted hazard ratios of acute pancreatitis by BMI group according to the presence or absence of type 2 diabetes. Adjusted for age, sex, alcohol consumption, smoking, type 2 diabetes, insulin administration, hypertension, dyslipidemia, chronic kidney disease, chronic pancreatitis, regular physical activity, and low income. * Statistically significant.

**Figure 2 jcm-11-05641-f002:**
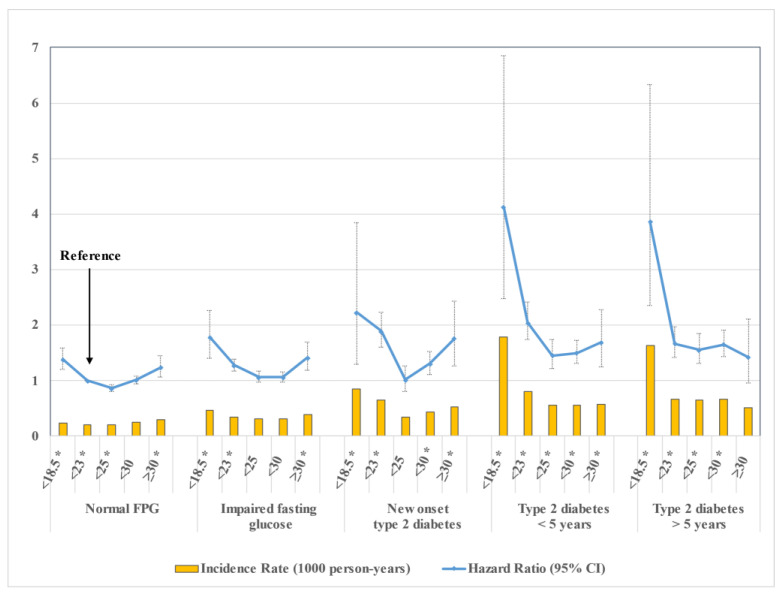
Incidence rates and adjusted hazard ratios of acute pancreatitis according to glycemic status and BMI category. Adjusted for age, sex, alcohol consumption, smoking, type 2 diabetes, insulin administration, hypertension, dyslipidemia, chronic kidney disease, chronic pancreatitis, regular physical activity, and low income. * Statistically significant.

**Figure 3 jcm-11-05641-f003:**
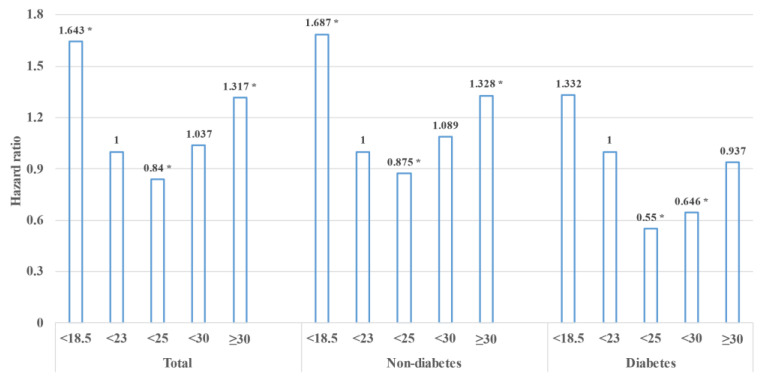
Adjusted hazard ratios of acute pancreatitis in subjects with constant BMI. Adjusted for age, sex, alcohol consumption, smoking, type 2 diabetes, insulin administration, hypertension, dyslipidemia, chronic kidney disease, chronic pancreatitis, regular physical activity, and low income. * Statistically significant.

**Table 1 jcm-11-05641-t001:** Baseline Characteristics.

	Patients without Diabetes and with a BMI < 18.5	Patients without Diabetes and with a BMI ≥ 18.5	Patients with Diabetes and with a BMI < 18.5	Patients with Diabetes and with a BMI ≥ 18.5	Absolute Standardized Difference (Max)
	(n = 140,516)	(n = 3,430,160)	(n = 4846)	(n = 336,974)	
Age, years	39.81 ± 16.29	46.45 ± 13.65	59.7 ± 15.74	57.26 ± 12.01	1.242
Male sex	45,163 (32.14)	1,878,366 (54.76)	2877 (59.37)	20,6840 (61.38)	0.613
BMI, kg/m^2^	17.57 ± 0.79	23.82 ± 3.29	17.43 ± 0.93	25.14 ± 3.2	3.272
Waist circumference, cm	66 ± 5.69	80.28 ± 9.1	70.27 ± 6.43	85.71 ± 8.77	2.666
Blood pressure, mmHg					
Systolic	113.33 ± 13.83	122.15 ± 14.76	123.08 ± 17.71	129.28 ± 15.82	1.073
Diastolic	71.12 ± 9.23	76.25 ± 9.99	75.67 ± 10.84	79.17 ± 10.24	0.826
Alcohol					0.257
Nondrinker	79,717 (56.73)	1,747,393 (50.94)	3022 (62.36)	191,982 (56.97)	
Mild	54,833 (39.02)	1,413,461 (41.21)	1408 (29.05)	111,272 (33.02)	
Heavy	5966 (4.25)	269,306 (7.85)	416 (8.58)	33,720 (10.01)	
Smoking					0.372
Nonsmoker	99,667 (70.93)	2,043,383 (59.57)	2577 (53.18)	187,838 (55.74)	
Former	9438 (6.72)	488,384 (14.24)	567 (11.7)	62010 (18.4)	
Current	31,411 (22.35)	898,393 (26.19)	1702 (35.12)	87,126 (25.86)	
Glucose, mg/dL	88.55 ± 10.71	92.73 ± 11.52	156.57 ± 68.39	146.51 ± 49.38	1.622
* Triglyceride, mg/dL	73.89 (73.71–74.07)	111.45 (111.38–111.51)	102.47 (100.84–104.13)	150.31 (150.01–150.61)	0.989
Total cholesterol, mg/dL	177.69 ± 35.41	195.78 ± 40.99	184.91 ± 47.85	197.77 ± 47.76	0.478
HDL cholesterol, mg/dL	64.31 ± 37	56.5 ± 32.51	60.01 ± 35.34	52.67 ± 32.94	0.332
LDL cholesterol, mg/dL	118.62 ± 381.44	122.13 ± 213.86	105.53 ± 151.43	113.18 ± 99.48	0.090
Hypertension	12,522 (8.91)	842,226 (24.55)	2004 (41.35)	201,813 (59.89)	1.272
Dyslipidemia	6880 (4.9)	565,078 (16.47)	1012 (20.88)	139,486 (41.39)	0.960
Insulin administration	0 (0)	0 (0)	290 (5.98)	12,687 (3.76)	0.354
Chronic kidney disease	7459 (5.31)	21,9381 (6.4)	665 (13.72)	42,446 (12.6)	0.290
Chronic pancreatitis	213 (0.15)	3460 (0.1)	46 (0.95)	874 (0.26)	0.136
Regular physical activity	13,099 (9.32)	61,3862 (17.9)	730 (15.06)	73,666 (21.86)	0.351
Low income	31,586 (22.48)	661,006 (19.27)	1118 (23.07)	69,326 (20.57)	0.093

* geometric means (95% confidence interval). Abbreviations: BMI, body mass index; HDL, high-density lipoprotein; LDL, low-density lipoprotein.

## Data Availability

The datasets generated during and/or analyzed during the current study are available from the corresponding author on reasonable request.

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
