# Peer review of "Underweight Is Associated with a Higher Risk of Acute Pancreatitis in Type 2 Diabetes: A Nationwide Cohort Study"

_jcm, 2022, doi:10.3390/jcm11195641_

Round 1

Reviewer 1 Report

I appreciate this study mostly as it uses a big amount of data and applies a considerable follow-up length. It is also analysed appropriately - research such as this can have big impact on medical science. 
On the other hand, my greatest concern is interest: we already know that underweight and obesity are associated with an increased risk of AP (the same stands for almost all diseases), this paper just demonstrates the same in diabetes. It is indeed the first one to examine this exact question (that i know of) and does so with big data and appropriate methods.
Some minor comments, questions:

In your methods you state: „Out of about ten million individuals that underwent the NHS program in 2009, 4,238,822 subjects, approximately 40%, were selected for this study. Why 40%? randomly or non-randomly?

I am assuming that you categorised participants based on their glycemic status and BMI in 2009. There was a subanalysis of patients with a constant BMI (Figure 3), was there any accounting for changes in glycemic status over time? If not, this is an important point for your limitations.

On Figure 2, the Hazard ratio line connecting between glycemic status categories is inappropriate. Within group, between BMI categories it is ok, but there is no continuous transition e.g. from normal fpg bmi 30 or greater to IFG bmi less than 18.5.

Star signs indicating statistically not significant results instead of statistically significant results in unusual, although it is probably because most results are significant.

The description of Figure 3 was left as the default by error, please correct this.

There is a previous meta-analysis of nearly 2,000,000 participants, in which, while underweight was associated with a higher risk of pancreatitis, obesity was more dominant (https://www.ncbi.nlm.nih.gov/pmc/articles/PMC7990844/ Fig 2). Looking at your Figure 1, in your analysis, being underweight is considerably more dominant. Any idea about the difference? Were your unadjusted HRs closer to this pattern? Or is it a difference in population?

Reviewer 2 Report

In this study, the authors report an association between body mass index (BMI) and acute pancreatitis in patients with type 2 diabetes. There is a large number of cases included in the analysis, and the paper is concise. However, there are some parts of the paper that must be made clearer.

Major points

1. Authors said “Out of about ten million individuals that underwent the NHS program in 2009, 4,238,822 subjects, approximately 40%, were selected for this study.” I couldn’t understand how 4,238,822 patients were chosen. 

2. It should be informative to have values of sensitivity, specificity, positive predictive value, and negative predictive value when using the electronic records to identify individuals with acute pancreatitis (outcome). Although the authors defined patients with acute pancreatitis as hospitalized cases in which ICD-10 code K85 labeled the primary diagnosis, there is no information on sensitivity, specificity, positive predictive value, or negative predictive value. Lacking this information, misclassification cannot be quantified and may be a major bias in effect estimation. A validation of the outcome definition and sensitivity analysis should be performed.

3. It should be informative to have values of sensitivity, specificity, positive predictive value, and negative predictive value when using these electronic records to identify individuals with other diseases or status (hypertension, dyslipidemia, chronic kidney disease, regular physical activity and low income).

4. Patients with chronic pancreatitis may develop emaciation from exocrine insufficiency or abnormal glucose metabolism from endocrine insufficiency. Furthermore, patients with chronic pancreatitis are generally more prone to acute pancreatitis. The primary concern of this study is that chronic pancreatitis may be a confounding factor, creating an apparent association between abnormal glucose metabolism and low body weight and acute pancreatitis. How were patients with chronic pancreatitis treated in this study?

5. When examining the differences between groups in table 1, the standardized difference should be evaluated instead of the p-value. 

Minor point

6. Is there any difference in the risk of acute pancreatitis depending on the type of diabetic medication? 

Round 2

Reviewer 2 Report

No additional comments.